# Genetic prediction of quantitative traits: a machine learner's guide focused on height

## Abstract

Machine learning and deep learning have been celebrating many successes in the application to biological problems, especially in the domain of protein folding. Another equally complex and important question has received relatively little attention by the machine learning community, namely the one of prediction of complex traits from genetics. Tackling this problem requires in-depth knowledge of the related genetics literature and awareness of various subtleties associated with genetic data. In this guide, we provide an overview for the machine learning community on current state of the art models and associated subtleties which need to be taken into consideration when developing new models for phenotype prediction. We use height as an example of a continuous-valued phenotype and provide an introduction to benchmark datasets, confounders, feature selection, and common metrics.

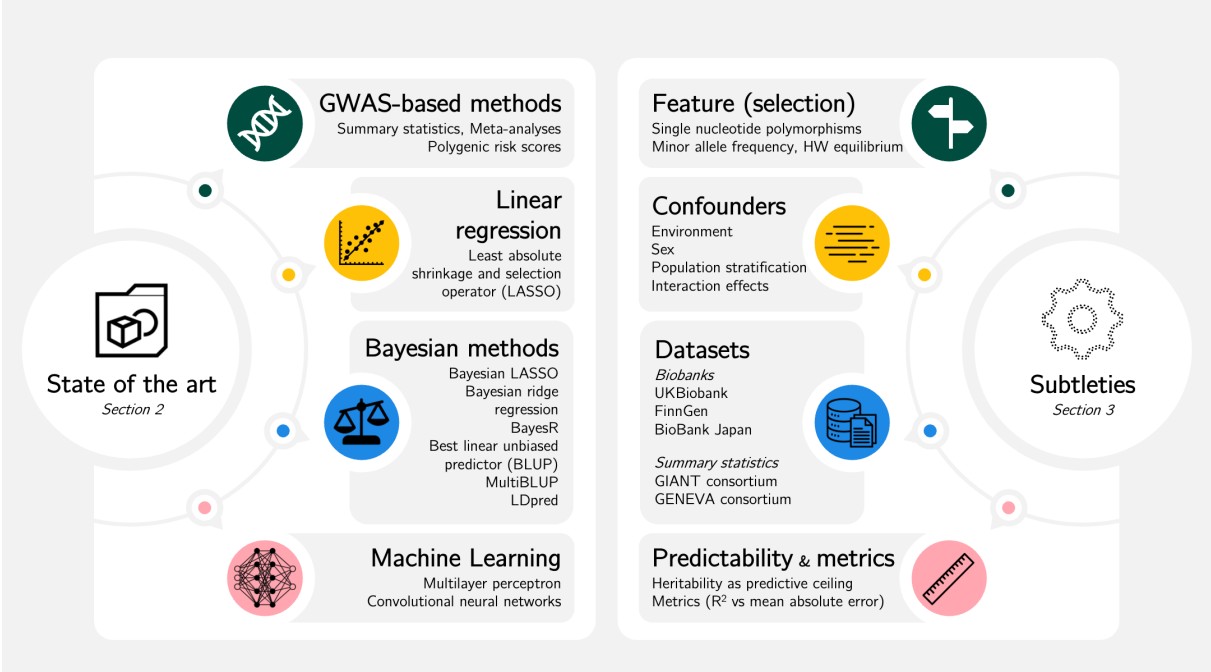

## 1 Introduction

Machine learning and especially deep learning has been tremendously successful in large number computational tasks such as computer vision (Voulodimos et al., 2018), natural language processing (Hirschberg & Manning, 2015), and, more recently, key computational biology problems such as protein structure prediction (Kuhlman & Bradley, 2019). With the advent of AlphaFold (Jumper et al., 2021) and ESMFold (Lin et al., 2022), it seems that one of they key challenges in structural biology, namely protein folding, is now

solved. Another prediction task of equal importance is *phenotype prediction.* Phenotype prediction can be loosely formulated as the (computational) task of predicting a binary or continuous phenotype from genetic information. A successful solution to this problem would have a disruptive influence on how personalised medicine is conducted. One could precisely estimate likelihoods of diseases such as diabetes (Kavakiotis et al., 2017; Padilla-Martínez et al., 2020) or cancers (Lee et al., 2019; Szulkin et al., 2015), body measurements such as height or body mass index (Robinson et al., 2015) or, potentially, even cognitive abilities (Allegrini et al., 2019). With precise and reliable predictions available, one has the ability to counteract unfavourable outcomes, i.e. very high risk of contracting certain types of cancers, from a very early stage (Lee et al., 2019). In this guide, we give an overview of the current state of the art models used to predict phenotypes and highlight various subtleties which machine learners need to consider when engaging with the difficult yet incredibly rewarding task of phenotype prediction.

Found in nearly all human cells, deoxyribonucleic acid (DNA) molecules contain the genetic blueprint to build and maintain living organisms. All human beings are 99.9% identical (Kruglyak & Nickerson, 2001) in their genetic makeup, yet there is an astonishing variety of individual differences in physical features (size, shape, skin colours), abilities (musical, athletic), and susceptibility to diseases. One of the most extensively studied human traits is height (Galton, 1886; Fisher, 1919), as it is considered an ideal model phenotype for studying continuous-valued complex polygenic (i.e. determined by multiple genes, contrary to Mendelian diseases which are determined by a single mutation) traits. Its prevalence in the literature can be attributed to multiple factors, namely (i) measuring height is easy, non-invasive and gives precise results, (ii) it is correlated to certain disease phenotypes (Raghavan et al., 2022), such as type II diabetes (D. et al., 2002) and prostate cancer (Zuccolo et al., 2008), and (iii) height is known to be a highly polygenic and heritable trait, i.e. an important proportion of variability in height observed in a population can be explained by genetic factors. Naturally, many phenotypes are only partially determined by the genetic makeup and the heritability of the given trait is defined as the amount of variability observed in the individual phenotype that can be explained by differences in genetic information. The heritability of height was first studied in twins and family cohorts (Verweij et al., 2011; Chatterjee et al., 1999; Silventoinen et al., 2003) where difference in heights between monozygotic and dizygotic pairs of twins along with non-zygotic siblings were compared. Such studies yielded an estimated heritability between 0.66 (specifically for women born before 1928) (Silventoinen et al., 2000) and 0.93 (based on another female cohort) (Silventoinen et al., 2003). This already gives a rough estimate that between 66% and 93% of the variance in height can be explained by genetic features alone. In machine learning terms, a model with a variance explained $R^2$ of these values amounts to perfect predictions. These results already highlight a fundamental difference between phenotype predictions and other types of common machine learning tasks: A perfect prediction ($R^2 = 1.0$, or equivalently 100% of the variance explained) is not possible, even in theory.

An obvious limitation of twin studies is that they do not have any predictive performance and to build predictive models, individual-level data is usually required. With the emergence of low cost genotyping solutions, such as genotyping arrays, individual-level genetic data have been collected in multiple large-scale biobanks across the world. It is often the single nucleotide polymorphisms (SNPs), which are, by definition, variations frequently observed in a given population, that are collected. SNPs hold the promise to better represent individual-level DNA variability, which could help us disentangle differences in the expression of phenotypes such as height. Thereby, heritability of height based on SNPs was approximated around 0.685 $\pm$ 0.004 (Ge et al., 2017). Again, put simply, a model which can explain $\sim 69\%$ of the variance in height could be considered a perfect model if it were not for *missing heritability* (Manolio et al., 2009).

By missing heritability we define the gap between heritability estimates from twin studies and heritability estimates from genetic data alone. A comparison of these estimates for height leaves up to 24.5% of supposedly genetic-related variability that remains unobserved. Recent work (Lello et al., 2018) claims that the problem of missing heritability based on common SNPs for height prediction was now solved using machine learning techniques. While this claim is engaging, it might not reflect the complexity of genetic-based phenotype prediction task.

The aim of this guide is to aggregate, discuss and refine the current approaches studying heritability and height prediction based on genetic information and make them accessible for the wider machine learning community. For that purpose, we examine the state-of-the-art practices and their corresponding results and

limitations. Following that inspection, we outline four subtleties when it comes to predicting height based on genetic information: (i) the features used and how to select them, (ii) the confounders to consider, (iii) the (benchmark) datasets available, and (iv) the predictability and metrics reported. Lastly, we conclude with recommendations on how to better report results for future studies, thereby improving comparability between studies, and argue that genetics might not be the only data needed to achieve accurate height prediction, moving towards the integration of both genetic and environmental data.

The guide is organised as follows. In Section 2, we review the most popular models used in height prediction and report quantitative results. In Section 3, we investigate the differences of the various reports and analyse the subtleties which need to be considered when reporting height prediction results. In particular, in Subsection 3.1, we discuss various methods of reducing the number of measured SNPs to a number manageable by statistical or machine learning models, i.e. we review currently used feature selection approaches. In Subsection 3.2, we discuss possible corrections for not measured non-environmental factors influencing prediction results. In Subsection 3.3 we discuss benchmark datasets which have been used, or could be used, in height prediction tasks and in Subsection 3.4 we study the importance of the various prediction metrics. We summarise our results in Section 4 and give recommendations for future studies.

## 2 State-of-the-art studies

There exists a large body of previous work in the genetics literature, which predicts height (among other continuous complex traits) from genetic data and population statistics. In this section, we summarise these approaches by the predictive models used and report their results. For comparability, all reported results in this section are expressed in the commonly-used variance explained ($R^2$) metric, if available.

### 2.1 Genome wide association study (GWAS)-based methods

A GWAS finds associations between individual SNPs and a phenotype of interest. For each SNP, a statistical test yields a p-value and an estimate of its effect size. SNPs that are the most strongly associated with the phenotype can further be combined in a unique score: the polygenic risk score (PRS). A PRS is obtained for each individual as a linear function of the estimated effect sizes, i.e. the regression coefficients, of the SNPs of interest, weighted by the strength of its association, i.e. the p-value, with the phenotype. An advantage of using PRS models is that they can be obtained using summary statistics instead of individual patients' genetic data, thereby protecting privacy rights of the study participants.

In this guide, we draw a clear distinction between "association testing", "heritability estimation" and "phenotype prediction". Association tests only search for SNPs that are statistically linked to a phenotype of interest, without aiming to predict the phenotype. Heritability estimation (e.g. Yang et al. (2015; 2010)) tries to calculate the variance explained by the selected SNPs without explicitly performing a prediction task. We excluded both such results for our discussion and focus only on models which can give a prediction of the phenotype for every individual.

Early works of the pre-biobank era usually used meta-analyses of published GWAS on smaller datasets and applied the unified findings to test sets. One such study (Lango Allen et al., 2010) performed a meta-analysis from 46 published databases, which identified 180 SNPs associated with adult height. The explained variance, evaluated on five external studies, was a sobering ∼10% of the phenotypic variance, although this may be due to stringent SNP significance inclusion criteria.

Another early study of Wood et al. (2014) develops a PRS for height by using a meta-analysis of published GWAS studies and prunes the SNPs based on p-value. It was shown that, based on the selection threshold, the most associated ∼2,000, ∼3,700 and ∼9,500 SNPs explained ∼21%, ∼24% and ∼29% of the height variance, respectively. These results were computed by evaluating the precomputed GWAS effect sizes of five different external validation datasets.

In a similar fashion, Paré et al. (2017) used the summary statistics for height of the Genetic Investigation of ANthropometric Traits (GIANT) consortium and evaluated the predictive performance of the PRS on the UK Biobank (Bycroft et al., 2018) and the Health and Retirement Study (HRS). The authors came to

a similar conclusion of a peak of ~22% variance explained in the UK Biobank and ~13% in the HRS. The decrease of the predictive power may be due to a different population structure of the test sets, i.e. the two populations might have different height distributions.

*Linkage disequilibrium* refers to the correlation between the features (SNPs) in the genetics literature and filtering out highly correlated SNPs is a standard preprocessing step. A classical PRS and a PRS using the pruning (on linkage disequilibrium) and thresholding (by p-value) approach, called P+T, was also computed in the seminal paper by Vilhjálmsson et al. (2015). They achieved a variance explained of 9.27% and 8.41% respectively on the GIANT dataset. When the population structure was taken into account explicitly, their estimates increased slightly to 12.05% and 11.46%. A P+T baseline was also reported in Márquez-Luna et al. (2021) with a variance explained of 34.81% on the UK Biobank data.

The interplay of the PRS and population structure was studied in Bitarello & Mathieson (2020); Martin et al. (2017); Duncan et al. (2019). In particular, in Bitarello & Mathieson (2020) a PRS was developed to include ancestry effects. However, no specific prediction task was considered.

It can also be shown that the predictive accuracy of the classical PRS improves drastically when parental height and genotype is known (You et al., 2021). The authors of this study used the UK Biobank and the Framingham Heart Study (FHS) to find parent-child relationships and compute an extended PRS based on the parents' PRS and their height. This method increased the variance explained by the PRS from ~73% to ~82%.

A combination of GWAS and methylation scores was studied in Shah et al. (2015), where they showed that, for height, methylation scores have very little predictive power and their maximum variance explained was ~19%.

## 2.2 Linear regression

Closely related to GWAS-based methods are linear-regression-based approaches, where, instead of a combined association testing and effect-size estimation, SNPs may be pre-selected and a linear model is trained on that potentially reduced feature set. Two recent studies (Lello et al., 2018; Qian et al., 2020) used least absolute shrinkage and selection operator (LASSO) regression applied to the UK Biobank SNPs.

In Lello et al. (2018), the authors used the 50,000 and 100,000 UK Biobank SNPs that were most strongly associated with height, according to GWAS. They then trained a simple LASSO model on these two sets of SNPS and showed that the maximum correlation between the predicted and actual height was 0.616 and 0.639 respectively when evaluated on a held-out test set from the UK Biobank. The performance on the Atherosclerosis Risk in Communities Study (ARIC) dataset was worse than UK Biobank internally, most likely due to imputation and difference in population structure.

By contrast, in Qian et al. (2020), the entire set of (unimputed) UK Biobank SNPs, consisting of 805'426 SNPs, was used. They then train a novel LASSO algorithm to achieve an explained variance of 69.9% on the UK Biobank test set.

## 2.3 Bayesian methods

Although LASSO regression could, in principle, be formulated as a Bayesian method, the above papers only consider the frequentist approach. In this subsection, we review papers which explicitly use a Bayesian regression method.

An early study of Makowsky et al. (2011) used a number of Bayesian models (Bayesian LASSO, Goddard-Hayes (Goddard, 2009), Yang (Visscher et al., 2010)) to predict height in the FHS dataset, where they achieved a maximum of 23.7% of explained variance with the Yang model.

In Kim et al. (2017), the authors investigated the effect of dataset scaling, in particular, the effect of the number of SNPs considered on prediction accuracy. They use Bayesian ridge regression and a model called BayesB (Meuwissen et al., 2001) to estimate the variance explained as a function of the number of SNPs on UK Biobank data. Their best estimates were achieved when the 50,000 SNPs (their maximum number

of SNPs studied) most strongly associated with height according to a GWAS on the train set were used and ranged from ~21% to ~23%, depending on the SNP selection procedure. The choice of model had a relatively minor impact.

In Berger et al. (2015), the authors used a Bayes A (Meuwissen et al., 2001) and a spike-slab (Ishwaran & Rao, 2005) model to predict height based on the "geneenvironment association studies (GENEVA)" dataset. They reported relatively poor prediction accuracy with correlations of 0.159 and 0.165 respectively.

In contrast to the previous studies, which used individual-level data, in Ge et al. (2019) only GWAS summary data was used in conjunction with a continuous-shrinkage prior Bayesian model. The explained variance of the models was ~28%.

Another summary-statistic-based method is SBayesR (Lloyd-Jones et al., 2019), which extends the popular BayesR model (Moser et al., 2015) to be used with GWAS summary statistics only. The best prediction accuracy was achieved when the 2.9 million most common SNPs in the UK Biobank dataset were used and the variance explained was 38.3%.

In Zeng & Zhou (2017) the authors considered a Dirichlet prior on the effect sizes of the SNPs. They developed a multiple regression model, including the SNPs and other covariates and used a Markov chain Mote Carlo (MCMC) and a variational Bayes (VB) method to fit the model. For height, the MCMC model performed best with a variance explained of 47.8% in the FHS dataset. The model performs as well as their BayesR (Moser et al., 2015) and reversible jump Markoc Chain Monte Carlo (MCMC) (Lee et al., 2008) baselines.

A specific heritability model was used as a prior in Zhang et al. (2021). There, the priors for the SNP effect sizes depend the heritability estimate for each SNP. The maximum achieved explained variance was 38% on the UK Biobank data with a training set size of 200,000 individuals.

### 2.3.1 Best linear unbiased predictor (BLUP) and its variants

The BLUP model is a common model in phenotype prediction, which has seen many extensions in the past decades (e.g. MulitBLUP (Speed & Balding, 2014) and GBLUP (de Los Campos et al., 2013)). Essentially, it is a (Bayesian) ridge regression model.

In Berger et al. (2015), two versions of BLUP (GBLUP and GBLUP-ldak) were used to obtain correlations between predicted and real height of 0.169 and 0.171 on the GENEVA dataset.

A weighted version of GBLUP was used in de Los Campos et al. (2013) to predict height in a combination of the FHS and GENEVA datasets. The best variance explained was 31.1% on the FHS dataset alone using the weighted GBLUP model.

MultiBLUP was used as a baseline in Zeng & Zhou (2017) on the FHS dataset. However, it performed worse than their MCMC Dirichlet prior model (exact number not published).

A similar explained variance to the previous approaches was presented in Xu (2017), which introduced a novel, alternative way to do cross-validation (CV). The authors used the BLUP model and 10-fold CV to obtain an explained variance of 30.6% in the FHS dataset. Their approach, which serves as an approximation to 10-fold CV, yielded 31.5% explained variance.

In Liang et al. (2020), the authors investigated the influence of SNP-density, haplotype, and dominance effects on the GBLUP model. Their best predictive model, which included all three effects and 380,000 SNPs, achieved a variance explained of 42.2%.

### 2.3.2 LDpred

LDpred is a popular method developed in Vilhjálmsson et al. (2015). It involved explicitly modelling the linkage disequilibrium which is usually done via an external validation dataset. There are two versions of LDpred, namely an exact version which relies on MCMC sampling and an approximate version (LDpred-inf) which, as the authors point out, is equivalent to GBLUP.

In the original paper, Vilhjálmsson et al. (2015) applied LDpred to the GIANT consortium data with the Mount Sinai Medical Center BioMe[1] as a validation cohort. Their variance explained was 10.14% and 9.06% for LDpred and LDpred-inf respectively. When population structure was taken into account, their estimates rose to 13.53% and 11.66%.

An LDpred baseline was also calculated in Lloyd-Jones et al. (2019). The study used the UK Biobank data and achieved a variance explained of 31.4%.

In Márquez-Luna et al. (2021), the authors develop some extensions to LDpred in which functional priors for the SNPs can be explicitly incorporated. On the UK Biobank dataset, the LDpred model achieves a variance explained of 38.20%. The LDpred extensions with functional priors, LDpred-funct and LDpred-funct-inf give 41.31% and 40.03% variance explained respectively.

### 2.4 Machine Learning

In this section, we briefly discuss non-linear machine learning approaches to predict height. All previous approaches, however sophisticated, assume a linear contribution of the SNPs to the phenotype, in our case height. It may be possible to obtain more precise estimates of height by considering and modelling epistasis, i.e. non-linear interactions between the SNPs.

In Bellot et al. (2018) the authors consider a variety of common deep learning algorithms such as multilayer perceptrons (MLP) and convolutional neural networks (CNN) to predict height. They used the UK Biobank dataset and selected the 10,000 and 50,000 SNPs most associated with height, based on a GWAS. They showed that the networks considered perform at most as well as the linear baselines (BayesR and Bayesian ridge regression) and obtained a correlation of ∼0.45. CNNs seem to perform slightly worse, with the exact correlation not being reported. An alternative SNP selection procedure based on windowed selection was also used, however, the performance deteriorated for all models.

An alternative approach was taken by Paré et al. (2017) where GWAS summary statistics (effect sizes) were first adjusted by a gradient-boosted tree correction and then a linear PRS was calculated. The GWAS effect sizes were taken from the GIANT dataset and training as well as test sets were from the UK Biobank. Their best prediction was a variance explained of 23.9%.

## 3 Subtleties of interpreting height prediction results

As can be seen from the previous section, the range of prediction accuracies appears to be very large, reaching from ∼9% to ∼80% in variance explained. In the following sections, we highlight subtleties in the different predictions and explain why a direct comparison of methods is often impossible.

### 3.1 Subtlety 1: Features and feature selection

All papers discussed in Section 2 used so-called single nucleotide polymorphisms (SNPs) as their feature set to predict height. In this subsection, we give an overview what SNPs are and how different preprocessing steps can lead to drastically different answers.

The human genome is encoded in the DNA, a polymer of roughly 3 billion nucleotide base-pairs (Nurk et al., 2022), built from 4 bases: adenine (A), thymine (T), cytosine (C) and guanine (G). From this polymer, one can extract several million SNPs (and Eric S. Lander et al., 2001; Hap, 2005). One SNP represents a variation in a unique base compared to a reference genome, which is present in at least 5% of the population studied (Hap, 2005). Those SNPs constitute the primary input for the prediction of polygenic traits, as they hold the promise of capturing variability in phenotype expression within a given population. Briefly, SNPs can be recoded into a compact vectorial form by counting how many copies of the diploid DNA are affected by a given polymorphism (0, 1, or 2) at each SNP location. This encoding strategy is referred to as additive encoding. Whilst the additive encoding is usually favoured, there exists other encoding schemes such as recessive/dominant or genotypic encoding (Mittag et al., 2015). In recent years, other approaches based

---

[1] https://icahn.mssm.edu/research/ipm/programs/biome-biobank

on whole genome sequencing (WGS) were made possible following technological improvements. However, they represent unique challenges in their computational design and in the resources required due to the very high-dimensional input from WGS. While non-genetic height prediction methods exist, such as prediction based on parents' height (Luo et al., 1998), or based on longitudinal population studies for benchmarking of children's growth (World Health Organization, 2009), we do not discuss these methods further.

### 3.1.1 Feature Selection

Despite confining our analysis to SNPs, which give the variation to the mean genotype, instead of WGS data, the resulting SNP matrix can still be too large to be handled by traditional machine learning techniques, especially considering the large sample size of biobanks reaching up to 500,000 participants. To this end, various feature selection methods have been proposed in the context of statistical genetics. Feature selection starts from the pre-processing of the dataset. SNPs can be selected based on missing calling rate, which is equivalent to filtering based on the proportion of missing data per SNP; the Hardy-Weinberg equilibrium (HWE) principle (Edwards, 2008), which tests for the allele frequencies in a population and where deviation from the equilibrium might be a sign of genotyping error (Hosking et al., 2004); or the minor allele frequency (MAF), which is based on the frequency of the second most prevalent allele in the population.

Traditional thresholds range between 1 and 10% for missing calling rate, between 5 and 0.1% for MAF and are often set to P> 0.0001 for HWE. Such thresholds were used in Lello et al. (2018); Bellot et al. (2018); Lango Allen et al. (2010); Vilhjálmsson et al. (2015); Shah et al. (2015).

Feature selection at the stage of pre-processing (usually called quality control, QC) can also be combined with GWAS. GWAS is the best known method for a univariate feature selection method. In GWAS, one fits a univariate linear regression to the genotype on the phenotype of interest, and determines if the effect size of each SNP estimated from the regression is significant or not. Conventionally, Bonferroni's correction is applied on the p-values estimated from each effect size to cope with the increase in false positives after multiple testing, and a p-value below $5*10^{-8}$ (Xu et al., 2014) is considered as an indication of the individual SNP being significantly associated with the phenotype of interest. Subsequent selection can be applied to only consider SNPs associated with the phenotype of interest. As an extension of the GWAS-based feature selection, another feature selection method, which we will refer to as the window-based approach, has been described more recently in Yengo et al. (2022). It consists of the selection of SNPs in a $2 \times 35$ kilo bases (kb) window around SNPs that are genome-wide significant for a phenotype of interest. The threshold of 35kb was chosen according to Wu et al. (2017), where they evaluated, based on a simulation study, that there is a probability of 80% for the causal SNPs, i.e. the SNPs directly causing the phenotype, to be located within this window. This method takes into account both linkage disequilibrium (LD) and the causal relevance of the alleles studied. Whilst this method seems to be of interest, it is not yet as developed as the GWAS-based feature selection and its superiority is yet to be established. There exist other approaches, which are largely knowledge-based. One might rely on large databases, such as DisGeNet (Piñero et al., 2019), which report SNPs known to be associated with a trait. Such SNPs can be chosen for the prediction task. The advantage of this method is that it usually results in a much smaller feature-space dimension and the known SNPs can stem from any source, either statistical association testing or clinical studies. On the other hand, however, one is limited to previous studies and new associations cannot be included. Network-based filters could, in principle, also be applied. Here, knowledge from protein-protein interaction networks are studied to identify SNPs located on genes coding for proteins involved in protein-protein interactions. This approach allows to explicitly take into account potential interactions between SNPs.

### 3.2 Subtlety 2: Confounders

The target phenotype, in our case height, is often influenced by non-genetic factors or genetic factors which cannot be inferred from the features. As we show in Subsection 3.4.3, correction for confounders can have a large impact on the metrics reported.

Although in this guide we focus on *genetic* prediction of complex human traits and height in particular, we briefly discuss the "environment" as a confounder to more accurate predictions. Human height can be influenced by many factors such as malnutrition as a child, severe or recurring diseases during the growth

phase, socioeconomic background (which can itself correlate strongly with, e.g. nutrition) and many other factors. These factors can to some extent be accounted for by establishing a predictive ceiling for height, i.e. the maximum possible variance explained by the genes (see Subsection 3.4.1).

More importantly, there are confounders which are to some extent neither purely genetic nor environmental or simply not covered by the input features. The most important example is sex. Although of genetic origin, in most studies the sex chromosome is excluded from the genetic data and, therefore, sex needs to be explicitly accounted for (see Table 1 in the Appendix). This can done either by adjusting males and females separately before any prediction task , or by including sex as a covariate in the model. Another such confounder is the year of birth, as populations especially in the western world tend to grow taller in the last centuries/decades (Max Roser & Ritchie, 2013). Since most data sources used in height prediction tasks involve participants born in an interval of years, and not a single year, this effect also needs to be taken into consideration, either by adding the year of birth as an explicit input feature, or by regressing age against the measured height for each participant.

A third category are population-stratification effects. Previous studies showed that ethnicity plays a important role in model performance and models trained on one ethnicity often do not generalise well when applied to a cohort of different ethnicity (Martin et al., 2017; Duncan et al., 2019; Yengo et al., 2022). The ethnicity confounder is often addressed by focusing on a subsample of the study cohort (usually the subpopulation of European origins due to data availability) . Even within cohorts unified for ethnicity, genetic kinship (family relations) and systematic genetic differences may remain, which would then often be addressed by regressing out (or adding as features) the genetic principle components computed from the population studied.

Although sometimes squared age effects, age-gender interactions and other confounders such as batch effects or geographic origin of the individual might also considered (e.g. Zaitlen et al. (2013)), the main confounders adjusted for in the literature are age and sex.

### 3.3 Subtlety 3: Datasets

In this subsection, we briefly discuss the different datasets available to perform prediction tasks, in particular, we classify the datasets as individual-level data or summary statistics data. The choice of dataset, specifically its size, may have a large influence on the accuracy of the predictions. As a general trend, more data tends to lead to more accurate predictions.

#### 3.3.1 Biobanks

Regardless of whether association testing, heritability estimates or (height) prediction tasks are performed, there is a need for large datasets in statistical genetics. In particular, this applies since the feature dimension is very large and many feature reduction techniques can only be performed with sufficient statistical power. Further, the development of complex models and reliable generalisation can only be achieved when sufficient data is available. To this end, many countries around the globe have established national biobanks. Currently, the available data is dominated by participants with European ancestries, including biobanks in Europe, North America and Australia and representing around 75% of the genotypes collected around the world (Yengo et al., 2022).

**UK Biobank (Bycroft et al., 2018):** One of the most widely used resources in height prediction is the UK Biobank (Ge et al., 2017; Kim et al., 2017; Lello et al., 2018; Sakaue et al., 2020). The UK Biobank consists of 500,000 participants who were between 40 and 69 years old at the time of recruitment. For most participants, genotype information of common variants is available and an increasing amount of rare genetic variants and exome data is being continuously released (Backman et al., 2021). Background information about the participant such as age and gender, baseline statistics, general health information, socioeconomic status, smoking status, etc. is available.

**FinnGen Biobank (Kurki et al., 2022):** The FinnGen biobank at the time of writing (12.07.22) consists of 392,000 entries[2] of combined genotype and health registry data. The total number of participants is about

---

[2]See https://www.finngen.fi/en

500,000. Due to the fact that every Finn can participate in FinnGen, the age range is expected to be larger than in the UK Biobank.

**Estonian Biobank:** The Estonian Biobank has currently close to 200,000 participants aged 18 or older[3]. It is a representative sample of the 1.3 million individuals in the Estonian population and it offers a wide range of genotypic and phenotypic data.

**BioBank Japan (Nagai et al., 2017):** The BioBank Japan is one of the largest and best established biobanks outside Europe. It consists of 260,000 participants [4] for which various types of genetic information is available, along with disease status for over 40 conditions and mortality.

**Framingham cohort (Cupples et al., 2003):** The Framingham cohort, originally designed to study risk factors linked to cardiovascular diseases, has extracted 550,000 SNPs for each of its 9,000 participants (Govindaraju et al., 2008). Whilst the primary aim of this cohort was not to study human height genetics, its size and quality made it a cohort of choice in the emergence of genetics studies, in a time where bigger national biobanks were not yet available (Lango Allen et al., 2010; Xu, 2017; Makowsky et al., 2011).

**1000 Genomes Project (Consortium, 2015):** The 1000 Genomes Project is an initiative promoted by the International Genome Sample Resource (IGSR), which was carried out between 2008 and 2015. During that period, over 2,500 genomes were collected across the world to serve as reference genomes for future analysis (Birney & Soranzo, 2015). To date, owing to its accessibility, it is still a major resource in the genetic field, included for height genetic prediction (Holland et al., 2020; Lloyd-Jones et al., 2019; Mostafavi et al., 2020).

**Others:** There are many smaller, more specific datasets such as the Atherosclerosis Risk in Communities Study (ARIC) including Icelandic, Dutch, and American populations (Gudbjartsson et al., 2008). There is also an Australian cohort dataset (Zhou et al., 2013). In the future, such datasets could also be used as external validation sets for methods developed on larger biobank-scale data.

### 3.3.2 Summary statistics

**Genetic Investigation of ANthropometric Traits (GIANT) consortium:** The GIANT consortium is an initiative which exists since the early 2010 and arose from the collaboration between 59 study groups [5]. As part of this collaborative effort, GWAS summary statistics were made available to research community for height, body mass index (BMI) and traits related to waist circumference. The GIANT consortium has already been the source for multiple studies developing PRS for height prediction, (Ge et al., 2019; Choi & O'Reilly, 2019).

**Gene Environment Association Studies (GENEVA) consortium:** The GENEVA consortium was founded in 2007 and currently includes 16 populations from diverse origins (European, Hispanic, Asian, African) across the United States. A more diverse set of phenotypes can be studied through this consortium, including addictions, oral health etc..

In general, we recommend to obtain access to at least two of the biobanks (one for training, one for testing) or other potential external test sets. There are many potential sources of covariate shifts in genetic data, with ethnicity being one of the most prominent, and careful evaluation of these effects is necessary when evaluating the generalisation performance of a model. One solution found was to restrict the analyses to individuals with Caucasian origins in cohorts where multiple ethnicity were reported (Lello et al., 2018; Bellot et al., 2018) (see also Table 1 in the Appendix). Whilst this approach tempers effects from population stratification, they also do not represent the general population anymore. This is especially true since no large scale biobanks

---

[3]See https://genomics.ut.ee/en/content/estonian-biobank
[4]See https://biobankjp.org/en/index.html#01
[5]See https://portals.broadinstitute.org/collaboration/giant/index.php/GIANT_Cohorts_and_Groups

for non-Caucasian population is currently available, with the exception of the Japanese biobank, which make specific genetic-based prediction in Hispanic, African or most Asian populations impossible.

Care needs to be taken also when selecting an external test set as the annotated SNPs between two biobanks might differ if they used different technologies. However, there exists standard in the annotation of SNPs such as the dbSNP Reference SNP, which most biobanks follow (Sherry, 2001). It could also be possible to find a direct correspondence between features in two biobanks based on the high correlation between certain pairs of SNPs (i.e. correspondence between SNPs could be found based on the study of LD).

Finally, whilst using summary statistics to develop models might be attractive as they are usually more easily accessible compared to individual-level SNP data, it should be noted that one should avoid using summary statistics from a certain population for model development and the same population for validation as it would introduce circularity in the model validation. This would yield artificially improved results. As an example, the GIANT consortium includes data from the Framingham cohort. As such, the Framingham cohort should not be used to validate a PRS which would have been developed based on the summary statistics from the GIANT consortium.

### 3.4 Subtlety 4: Predictability and metrics

In this section, we explain the subtleties of the overall predictability of the trait and also the metrics to estimate the prediction accuracies.

#### 3.4.1 Heritability as a predictive ceiling

To correctly evaluate the accuracy of any prediction of complex traits, or, indeed, any machine learning algorithm, one needs to establish theoretical bounds on the "predictability" of such traits. Traditionally, machine learning, and deep learning algorithms in particular, excel at tasks such as image or text classifications from test data with prediction accuracies often reaching over 90% (Zhai et al., 2021; Yang et al., 2019). In such tasks, however, the target predictions are noiseless with the features (pixels or sentences) containing all the necessary information for theoretically perfect prediction. In the situation of predicting polygenic traits such as height from genetic data, the situation is different as genes only determine part of the variation of the trait in a population with the remaining part being influenced by the environment (Visscher et al., 2008; Silventoinen, 2003). Therefore, before evaluating any output from a prediction algorithm one needs to establish the maximum predictability, in the genetic context called *heritability*.

Heritability defines the extent to which the variation of a polygenic trait is determined by the genes. It can be defined as the variance explained by all genetic factors and is highly dependent on the population studied. Recent work has found that heritability of height varies with birth country (Silventoinen et al., 2003), year of birth (Silventoinen et al., 2000), within subpopulations (Silventoinen et al., 2000; 2001; Tropf et al., 2017) and economic affluence of the study group (Silventoinen et al., 2000). Despite these issues, one can estimate heritability for the specific study group in question by pedigree analysis (Lange et al., 1976; Wang et al., 2011) before proceeding to any prediction tasks. In fact, there is no such thing as one "heritability", one commonly distinguishes between *broad sense heritability*, *narrow sense heritability* (Jacquard, 1983), and SNP heritability (Yang et al., 2017).

To understand the differences between broad sense and narrow sense we first investigate models of (variation of) traits. Any phenotype $y$ can be modelled by

$$y = f(g, e), \tag{1}$$

where $f(\cdot, \cdot)$ is a potentially nonlinear function of the genotype $g$ and the environment $e$ of an individual. The total variation of $y$ be expressed by

$$\sigma_y^2 = \sigma_g^2 + \sigma_e^2 + \sigma_{g,e}^2 + \sigma_{g \times e}^2, \tag{2}$$

where $\sigma_y^2, \sigma_g^2, \sigma_e^2$ are the variance of the phenotype, genotype and environment respectively. The terms $\sigma_{g,e}^2$ and $\sigma_{g \times e}^2$ account for the genotype-environment covariance and the genotype-environment interactions.

Because these terms cannot be estimated (Visscher et al., 2008) a simplified model is given by

$$\sigma_y^2 = \sigma_g^2 + \sigma_e^2. \tag{3}$$

Notice that this assumption is captured by the predictive model via a decoupling of $f(g, e)$ into $f(g) + \tilde{f}(e)$, where the covariance $\text{cov}(f, \tilde{f}) = 0$.

Broad sense heritability is defined as the total influence of the genetics on the variation of the trait and can be expressed by

$$H^2 = \frac{\sigma_g^2}{\sigma_y^2} = \frac{\sigma_g^2}{\sigma_g^2 + \sigma_e^2}. \tag{4}$$

Narrow sense heritability uses a decomposition of the variance $\sigma_g^2$ into additive, dominance and epistatic influences of the genes on the phenotype,

$$\sigma_g^2 = \sigma_a^2 + \sigma_d^2 + \sigma_{\text{epi}}^2, \tag{5}$$

where the additive variance $\sigma_a^2$ assumes independent action of each gene (i.e. $f(\cdot)$ is a linear function), dominance variance $\sigma_d^2$ accounts for the fact that a modification on only one allele may not have an effect (or, vice versa be completely dominant) and the epistatic variance $\sigma_{\text{epi}}^2$ captures the interaction of SNPs at different loci in the genome. Narrow sense heritability is defined as the total *additive* effect on the phenotypic variance,

$$h^2 = \frac{\sigma_a^2}{\sigma_y^2} = \frac{\sigma_a^2}{\sigma_g^2 + \sigma_e^2}. \tag{6}$$

Note that $h^2 \leq H^2 \leq 1$.

The SNP heritability ($h_{\text{SNP}}$) is defined as the proportion of variance explained by any subset of SNPs at hand (Yang et al., 2017). Often this is estimated from a dataset using a genomic restricted maximum likelihood (GREML) model (Yang et al., 2010).

Most estimates of height heritability range from 0.6 to 0.9 with a consensus in the surveyed literature that "the heritability of human height is around 0.8" (Visscher et al., 2006). Although the heritability estimates range widely, we believe that this may be due to different definitions of heritability and different phenotype corrections used (see Subsection 3.2 for a discussion). In order to judge the performance of any machine learning algorithm predicting a complex trait, it is crucial to start with an estimate of heritability as the predictive ceiling. One can obtain such an estimate from twin studies (Silventoinen, 2003; Silventoinen et al., 2001) or, as mentioned above, from the given SNPs using GREML. Although this heritability estimate may not be unbiased, it is a first step to making genomic predictions more comparable, especially if an appropriate metric is chosen.

Many current papers use simulation studies to fix the heritability for the simulated data and benchmark their algorithms against this parameter in order to find the missing heritability (difference between $h^2$ and $R^2$) produced (Choi & O'Reilly, 2019; Mostafavi et al., 2020; Vilhjálmsson et al., 2015; Márquez-Luna et al., 2021; de Los Campos et al., 2013; Holland et al., 2020; Ge et al., 2019). Most studies reviewed in this paper also obtain a heritability estimate for their dataset (see Table 1 in the Appendix). Note that we are purposefully vague as to which heritability is given as this depends on the type of model used.

### 3.4.2 Prediction metrics

In this subsection, we discuss common prediction metrics and their subtleties. The popular variance explained $R^2$ metric is discussed in particular detail due to its prevalence in the literature.

**Variance Explained ($R^2$)** The $R^2$ statistic is also called coefficient of determination or variance explained. It is determined by subtracting the ratio of the sum squared regression error $SS_{residual} = \sum_i^n (y_i - \hat{y}_i)^2$ and the sum squared total variation in the whole dataset $SS_{residual} = \sum_i^n (y_i - \overline{y})^2$, from one:

$$R^2 = 1 - \frac{\sum_i^n (y_i - \hat{y}_i)^2}{\sum_i^n (y_i - \overline{y})^2} \tag{7}$$

An $R^2$ value of one indicates a perfect prediction; zero indicates that the prediction is just as good as predicting the average value over the whole dataset and a negative value indicates a prediction worse than predicting the average value over the whole dataset. In genomic regression tasks, the variance explained is always upper bounded by the heritability (SNP or otherwise). Therefore, it would make sense to use the ratio of $R^2$ to a heritability estimate as an indicator of model performance. However, this is rarely done in the surveyed literature. Furthermore, the value of the $R^2$ metric depends heavily on how exactly it is calculated (see Subsection 3.4.3) and, therefore, it can be difficult to compare these metrics at face value. In particular, one can calculate the *partial* $R^2$ value which only takes confounder-corrected phenotypes into account or the total $R^2$ which uses the raw phenotypes.

**Mean squared error (MSE)**  The mean squared error states the average squared difference between estimated and true value:

$$MSE = \frac{1}{n} \sum_{i}^{n} \left(y_i - \hat{y}_i\right)^2 \tag{8}$$

The MSE is a popular metric in machine learning. It is always positive and takes a value of zero when predictions are perfect. There is no upper limit on the MSE value, it is therefore task dependent to gauge what should be considered a favourable MSE value and comparison of MSE values between tasks is not always possible. MSE is expressed in the squared units of the outcome to be predicted and is particularly affected by outliers which lead to bigger differences between predicted and true value. In the case of height, the unit of the MSE would be centimetres squared ($cm^2$), which may not be very informative. The *root* mean squared error (RMSE) is again measured in centimetres ($cm$) and it would estimate the standard deviation of the predictions.

**Mean absolute error (MAE)**  The MAE represents the average difference between the value predicted and true value:

$$MAE = \frac{1}{n} \sum_{i}^{n} |y_i - \hat{y}_i| \tag{9}$$

Similarly as for MSE, the MAE is always positive and a value of zero represents a perfect prediction over the entire test set. By contrast to the MSE, MAE is directly expressed in the same units as the outcome to be predicted. We argue that the MAE might be the most useful metric in reporting quantitative trait predictions, in particular height, as it gives a concrete and intuitive answer to the question "If I were to predict your height based on your genetic information then how far is my prediction (on average) away from your real height?".

**Connection between $R^2$ and MSE**  The described metrics are related to one another by the following formula:

$$R^2 = 1 - \frac{SS_{residual}}{SS_{total}} = 1 - \frac{\sum_{i}^{n} \left(y_i - \hat{y}_i\right)^2}{\sum_{i}^{n} \left(y_i - \overline{y}\right)^2} = 1 - \frac{n\,MSE}{n\,\sigma^2} = 1 - \frac{MSE}{\sigma^2},$$

where $\sigma^2$ is the total variance of the phenotype in the dataset. $SS_{residual}$ refers to the residual sum of squares and can also be interpreted as the sum of squares not explained by the model predictions, as the following holds true:

$$SS_{total} = SS_{explained} + SS_{residual}$$

This representation indicates that both the MSE as well as the overall variance in the data influence the value of $R^2$.

### 3.4.3   Simulation studies reveal metrics behaviour

We simulate two datasets with 2,000 instances each. The first simulated dataset is sampled from a single normal distribution with mean $\mu$ equal an overall mean height of 165 cm ((a) in Figure 1). The second is composed of 2 samples (of 1,000-instances each) from two normal distribution representing both genders with the ((b) and (c) in Figure 1). The average male and female height were set to 171 cm and 159 cm

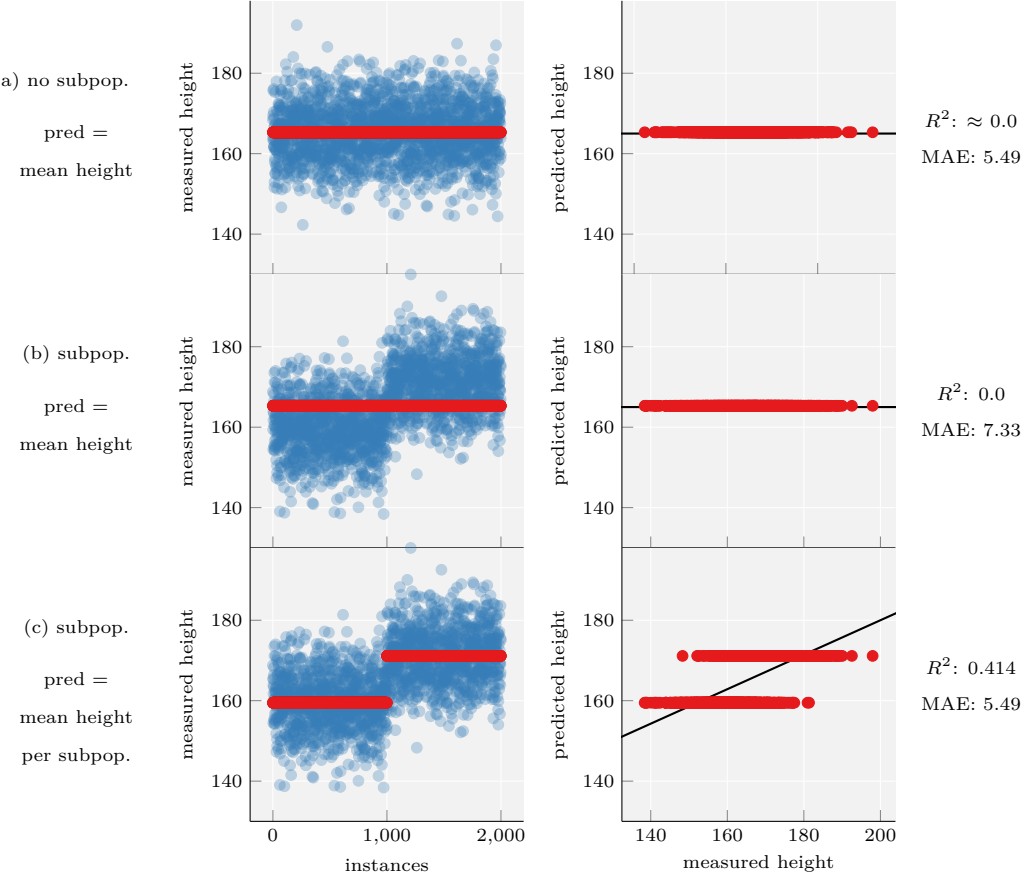

Figure 1: $R^2$ and MAE behaviour in population with and without subpopulations, simulating both genders. (a) Population drawn from the same normal distribution (overall mean height 165cm) with height prediction by taking population mean, leading to a $R^2$ of $\approx 0.0$ and MAE of 5.49. (b) Population drawn from two normal distributions (male mean height 171cm, female mean height 159cm) with height prediction by taking population mean, leading to a $R^2$ of 0.0 and MAE of 7.33. (c) Population drawn from two normal distributions (male mean height 171cm, female mean height 159cm) with height prediction by taking mean per subpopulation, leading to a $R^2$ of 0.414 and MAE of 5.49.

respectively (Max Roser & Ritchie, 2013). Simulating two distinct scenarios allows us to illustrate an homogeneous population and a population with subpopulations, respectively. In all sampling experiments, we drew samples from the normal distributions with a sigma value of seven (Hill, 2012). We depict the behaviour of $R^2$ and MAE in simulated height prediction tasks in Figure 1. In the first two cases, we set the predicted height as the overall average height in the predicted dataset ((a) and (b) in Figure 1); in the third case predicted height is set to the mean height per subpopulation (c). $R^2$ shows values approximately 0.0 for both cases where the overall height in the data is predicted (a)(b), even though the prediction are clearly further off in the case with subpopulations (b). The MAE indicates this difference in the fit through a larger values in (b). However, the most unintuitive behaviour is observed when comparing (a) and (c). In both cases, the relative goodness of fit of true and predicted height is the same. This is indicated by the MAE having the exact same value for both predictions. The $R^2$ however indicates a relatively high value in the case of subpopulations (c).

### 3.4.4 Discussion of the simulation study

To summarise this subsection, we briefly discuss the findings of the simulation study and give suggestions to ensure better comparability of the results. If we only predict average height (the simplest scenario), then

the $R^2$ value is zero in both cases. However, the MAE clearly points out worse *overall* predictions. Likewise, the $R^2$ value is automatically inflated, when predictions are made on a corrected phenotype and the $R^2$ is calculated. The prediction model used is *not* a more complex model, the population stratification is simply taken into account (e.g. by z-scoring male/female heights).

Therefore, we argue that for quantitative traits and better comparability across the literature, it is crucial to not only report $R^2$ but also the MAE systematically. One possible drawback of the MAE is that, unlike the $R^2$, the MAE has units, namely the units of the original phenotype. This leads to less comparable results across phenotypes. Additionally, when predicting a quantitative phenotype, the average error is also usually the most clinically relevant quantity (e.g. outliers would be of more interest in predicting the blood cholesterol levels).

The total $R^2$ value is also sensitive to the effect size of the population stratification, e.g. when two sub-populations are present with vastly different means. Hence, we recommend to also always report the average height in the dataset and the average heights of the subpopulations to be able to estimate the effect size of the inflation of $R^2$.

## 4    Discussion and recommendations

Starting in the early 2000s with the rapid development of genotyping arrays and sequencing methods, genetic research holds the promise to inform the human physiopathology at a whole new level: the level of our DNA. The attention was initially directed towards GWAS, where researchers tried to find genetic components associated with certain phenotypes, diseases in particular. It soon came to light that no single genetic component, with the exception of Mendelian traits, would likely explain the diversity in phenotypes and diseases presentation. That realisation led to the development of risk scores, such as PRS, representing one's genetic susceptibility to acquire a disease. In order to bring the use of the genetic information one step further, research has involved into genetic-based prediction of phenotypes. Such predictions are far more complex than the previous methods used for genetic data, as it appears that genetics is highly connected to the environment in which it is expressed. In this guide, we introduced the complex topic of phenotype prediction to the machine learning community. We bridge the gap between the fields of statistical genetics, which is largely concerned with finding associated of phenotypes and genetic variations and linear models of phenotype prediction, and machine learning, which has the potential to revolutionise phenotype prediction with complex, non-linear models. Throughout our analysis, we intended to document the current limitations of genetic-based phenotype prediction, taking height as our illustrative example, and provide potential guidelines in order to alleviate the complexity of such a prediction task. As described in Subsection 3.1, SNPs are the main source of data for genetic prediction. We described commonly used feature selection procedures for SNPs and essential quality control measures. Whilst the idea of taking advantage of SNPs present in a certain proportion of the study population (usually around 1 to 5%) remains appealing, it clearly does not represent the entire diversity of our genomes. Thus, current research enlarges the pool of features considered to include so-called rare variants (present in less than 1% of the population studied), epigenetic variations, copy number variants, etc. Enriching the feature space has the potential to better represent the complexity of the genetic landscape at the individual level and, therefore, provide more leverage for machine learning approaches.

Those emerging genetic modalities are currently not included in most estimations made for heritability of traits, which constitutes the predictive ceiling. It should be considered that, along with a precise definition of the type of heritability estimated, future research should include those rare variants in their estimation of heritability. Similarly, if heritability can be used as a metric for estimating the accuracy of the model's prediction, it should not be studied alone, but rather in addition to other metrics. Currently, as described in the Subsection 3.4, $R^2$ is the dominating metric reported in the literature. However, the $R^2$ depends both on the number of features considered (Mittlböck & Schemper, 1996) and the presence of the sub-populations (Ranganathan & Aggarwal, 2016). Whilst the first point is now easily mitigated by the increasing size of biobanks (i.e. we have access to better benchmark datasets), the presence of sex-related sub-populations needs to be accounted for when predicting phenotypes such as height, as illustrated in Fig 1c. We suggest to favour the MAE and RMSE for the model's accuracy evaluation, or to train different models for each

sub-population. This second option holds the advantage to align with what has long be done in the twin studies, the initial source of height heritability estimates.

Twin studies similarly revealed differences in heritability based on ethnicities (Silventoinen et al., 2003). Currently, the data sources available are largely restricted to white Caucasian populations, thus restraining the interpretation and validation of prediction models to this ethnic group. It should be noted the "white Caucasian" is in itself a rather diverse group, ranging from Nordic ancestry to Mediterranean populations. Significant efforts should be directed towards the collection of data from more diverse ethnic groups, such as African and Asian populations. This is of even greater importance when facing deployment of any machine learning model in clinical practice. To assure fair and accountable results, a model should not only be developed based on a representative sample of the target population, but also be validated on an external equally-representative cohort, before being more largely used. Recent failures of artificial intelligence models to obtain fair results when faced with underrepresented populations (Feuerriegel et al., 2020) demonstrate the subtlety of this task and need to be taken into account, especially in such a high-stakes task as predicting disease phenotypes. Finally, after studying the current landscape of the literature on genetic-based height prediction, it appears that, even for a benchmark phenotype which is largely determined by the SNP features considered (i.e. the heritability is high), genetic information alone might not be sufficient to achieve sufficiently accurate predictions. Thus, the next breakthrough in using genetic data could be based on its integration along with non-genetic information, such as environmental factors. Taken together, they could better leverage the high amount of data collected nowadays to achieve more precise and meaningful phenotype prediction and, thus provide another success story for machine learning.

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

## 5 Appendix

Table 1: An overview of the reviewed papers. We report the details relevant for all the "subtleties" concerned in this review. Each reviewed paper may study multiple models, but we only highlight the best model and its performance. Dataset abbreviations: UKB = UK Biobank, FHS = Framingham Heart Study, GENEVA = Gene Environment Association Studies, GIANT = Genetic Investigation of ANthropometric Traits; MAF = minor allele frequency

| Paper | Datasets | # SNPs and Participants | Feature selection | Best model | Best result | Confounder correction | Heritability estimate |
|---|---|---|---|---|---|---|---|
| Lango Allen et al. Lango Allen et al. (2010) | Meta GWAS | ~3M and 133,653 | GWAS | PRS | $R^2 = \sim 0.1$ | adjusted; age, sex, "appropriate covariates" | yes |
| Wood et al. Wood et al. (2014) | Meta GWAS | ~2.5M and 253,288 | GCTA-COJO | PRS | $R^2 = 0.29$ | adjusted; top 20 PCs | yes |
| Paré et al. Paré et al. (2017) | GIANT/UKB | ~2M (GIANT), various UKB | None | GrabLD | $R^2 = 0.239$ | adjusted; age, sex, top 15 PCs, Europeans | yes |
| Vilhjálmsson et al. Vilhjálmsson et al. (2015) | GIANT, Mount Sinai | ~539k and 133,653 + 2,013 | dataset overlap | LDpred | $R^2 = 0.14$ | adjusted; top 5 PCs | yes |
| Márquez-Luna et al. Márquez-Luna et al. (2021) | UKB, 23andMe | ~6M and 408,092 | None (UKB) | LDpred-funct | $R^2 = 0.41$ | No (UKB) | yes |
| You et al. You et al. (2021) | UKB, FHS | 3,290 and 1,017 (UKB), 444 (FHS) | GWAS | PRS + parental heights and PRSs | $R^2 = 0.82$ (UKB) | covariates; sex, age, top 40 PCs and top 40PCs of parents, Europeans | no |
| Shah et al. Shah et al. (2015) | Lothian Birth Cohorts (LBC), LifeLinesDEEP, GIANT | Not reported and 1,641 (LBC), 752 (LifeLines) | MethylationWAS | PRS + MRS + PRS×MRS | $R^2 = 0.19$ | adjusted; age,sex,generation | no |
| Liang et al. Liang et al. (2020) | FHS | ~500k and 7,565 | MAF and selecting every $x^{th}$ SNP | GBLUP | $R^2 = 0.42$ | covariate; sex,age, unrelated Caucasian | yes |
| Makowsky et al. Makowsky et al. (2011) | FHS | max ~400k and 7,565 | MAF | Bayesian | $R^2 = 0.36$ | adjusted; sex | yes |
| Berger et al Berger et al. (2015) | GENEVA | ~780k and 5,961 | MAF and QC | GBLUP-ldak | $R = 0.17$ | adjusted; sex, age, case/control, unrelated Caucasian | yes |
| Xu Xu (2017) | FHS | ~500k and 6,161 | None | BLUP | $R^2 = 0.31$ | covariate; sex, generation | yes |
| Kim et al. Kim et al. (2017) | UKB | max 50k and 102,221 | GWAS | Bayesian | $R^2 = 0.24$ | adjusted; sex, age, unrelated Caucasian | yes |
| Ge et al. Ge et al. (2019) | Partners Biobank | 750,888 and 3957 | GWAS | PRS-CS | $R^2 \sim 0.28$ | adjusted; age, sex, top 10 PCs, European | yes |
| Lloyd-Jones et al. Lloyd-Jones et al. (2019) | UKB | ~2.9M and 347,106 | None | SBayesR | $R^2 = 0.383$ | adjusted; sex, age, top 10 PCs, unrelated European | yes |
| Zeng et al. Zeng & Zhou (2017) | FHS | ~388k and 6,950 | None | BayesR, rjMCMC | $R^2 \sim 0.478$ | adjusted; quantile normalized | yes |

Table 1: An overview of the reviewed papers. We report the details relevant for all the "subtleties" concerned in this review. Each reviewed paper may study multiple models, but we only highlight the best model and its performance. Dataset abbreviations: UKB = UK Biobank, FHS = Framingham Heart Study, GENEVA = Gene Environment Association Studies, GIANT = Genetic Investigation of ANthropometric Traits; MAF = minor allele frequency

| Paper | Datasets | # SNPs and Participants | Feature selection | Best model | Best result | Confounder correction | Heritability estimate |
|---|---|---|---|---|---|---|---|
| Zhang et al. Zhang et al. (2021) | UKB | ~630k and 220,00 | None | BayesR + BLD-LDAK | $R^2 = 0.384$ | adjusted; sex,age,Townsend deprivation index, top 10 PCs, unrelated white British | yes |
| Lello et al. Lello et al. (2018) | UKB | top 100k and 488,371 | GWAS | LASSO | $R = 0.64$ | adjusted: sex, age, unrelated European | no |
| Qian et al. Qian et al. (2020) | UKB | ~800k and 337,199 | None | LASSO | $R^2 = 0.70$ | covariate; sex, age, top 10 PCs, unrelated white British | no |
| De los Campos et al. de Los Campos et al. (2013) | FHS, GENEVA | 400k and 5,800 | Overleap between datasets | weighted GBLUP | $R^2 = 0.311$ | adjusted; sex, age, Caucasian | yes |
| Bellot et al. Bellot et al. (2018) | UKB | top 50k and 102,221 | GWAS | MLP and LASSO | $R \sim 0.45$ | adjusted; age, sex, center, top 10 PCs | no |

