# OpenReview forum: "Genetic prediction of quantitative traits: a machine learner's guide focused on height"
_TMLR — Withdrawn by Authors_

### Review · Reviewer_28Zi · 2023-04-17

**Summary Of Contributions:**

The paper argues that genetics might not be the only data needed to achieve accurate height prediction, moving towards to the integration of both genetic and environmental data. One of the ideas is to bridge the gap between statistical genetics and machine learning (and to apply complex, non-linear methods).

**Audience:**

No

**Claims And Evidence:**

Yes

**Requested Changes:**

I do not request any changes.
I guess that the current contribution is out of scope of TMLR.

**Strengths And Weaknesses:**

Strengths. The paper presents the rich literature on height prediction: a number of statistical and some machine learning methods are mentioned. Possible confounders are discussed, prediction metrics are provided. The limitations of genetic-based phenotype prediction are documented.

Weaknesses. The paper is focused on a specific task: height prediction; the findings are not original with respect to the machine learning field. The current contribution does not really consider any machine learning method. The paper proposes to favour MAE and RMSE in the applications of height prediction.

---

### Review · Reviewer_2BN5 · 2023-05-06

**Summary Of Contributions:**

The authors describe the current state of the field in understanding the link between genetic variation on height with mathematical models. The authors describe the caveats of approaches in statistical genetics, as well different approaches in the literature to predict height from genetic variation.

**Audience:**

No

**Broader Impact Concerns:**

I think the discussion of the ethical impact of deploying PRS models to be informative, helpful, and well written.

**Claims And Evidence:**

Yes

**Requested Changes:**

As noted previously, there are a number of terms that need to be more fully described to make this work accessible to a machine learning audience. While I realize these terms may be very common in the statistical/human genetics community, they are likely new to the machine learning community:
Summary statistics - What columns are typically reported, and what do each of those columns mean?
Sequencing technologies and techniques - In the paper, you note: “Care needs to be taken also when selecting an external test set as the annotated SNPs between two biobanks might differ if they used different technologies.” The machine learning audience is likely not aware of how DNA is sequenced, and that biases can occur with different technologies.
Hardy-Weinberg equilibrium - there was a brief explanation, but more discussion would be helpful.
“Thus, current research enlarges the pool of features considered to include so-called rare variants (present in less than 1% of the population studied), epigenetic variations, copy number variants, etc.” To the machine learning community that has no biology background, what are each of these different kinds of variants? How do they differ from SNPs? Why aren’t they included in GWAS/PRS calculations?
Why aren’t alleles on the sex chromosomes used for height GWAS? Can you link to studies discussing this, or potentially elaborate further?

I found the first introduction paragraph to be out of place with the rest of the work. The first paragraph touts the importance of deep learning, but the actual content of the paper barely mentions it.

Heritability (and associated metrics with respect to height) are mentioned in the introduction. It’d be great to have the mathematical definition of heritability more to the front.

“Briefly, SNPs can be recoded into a compact vectorial form by counting how many copies of the diploid DNA are affected by a given polymorphism (0, 1, or 2) at each SNP location.” – I don’t think “affected” is the right word here. Moreover, to the ML community, this encoding scheme of the number of non-reference alleles needs to be better defined. Moreover, why not define an encoding scheme where the actual alleles present are featurized (i.e. if C is major allele, why not C/A vs C/C vs C/T?

I realize this would be a lot of work, but much of the subtleties the authors discuss across datasets and methods could be addressed by actually downloading the data and reimplementing the methods in a standardized fashion. This standardized analysis would be a valuable resource to compare methods, instead of rehashing what exists in the literature with the number of caveats presented.

“Network-based filters could, in principle, also be applied. Here, knowledge from protein-protein interaction networks are studied to identify SNPs located on genes coding for proteins involved in protein-protein interactions.” Is there a reference here?

“​​There are many potential sources of covariate shifts in genetic data, with ethnicity being one of the most prominent…” Instead of ethnicity, do you mean ancestral origin?

As noted in the strengths and weaknesses section, I don’t think that the simulation study is informative, and I think it adds very little to the work.

It would also be nice if the table at the very end could be summarized into a smaller table in the main text.

In section 3.1, why is a minor allele frequency of 0.05 used from Hap et al?


**Strengths And Weaknesses:**

Strengths

I think the statistical definition of heritability is very useful and well written. I think this is a great thing to review with the machine learning community. I think it is great to make a link between this and those metrics used by the machine learning community, because they are a common language and framework to describe variation.

I think the discussion of confounders is useful (Section 3.2). I think it is good to break down all the aspects that could have an effect on human height, and how the field controls for those affects to isolate the genetic component.

I think the discussion of the difficulty of deploying PRSs in the real world to be very valuable. I don’t think the broader machine learning community is fully aware of the ethical considerations of when and how these models work, as well as how people would be impacted by them

Weaknesses

Generally, I do not think this paper is accessible to a machine learning audience, as stated in the title of the paper. There are many discussed topics in statistical genetics that are simply glossed over that would require quite a deal of explanation for someone with no biology or genetics background. Please see the requested changes below for items that need more discussion.

I also found that the authors limited themselves by only focusing on papers that predicted height. There must be other methods and approaches that have been published in which height was not the focus.

I found the simulation study to be weak and uninformative. The entire work is about better linking the effect sizes of variants across the genome to a phenotype of interest. In this simulation study, genetic variants and their effect sizes aren’t even included in the simulation–only the phenotype is simulated. Moreover, how does this simulation address any aspect of heritability?

To me, the real question is why non-linear, machine learning-based methods do not perform better than linear mixed models? Why is it that only variants of linear regression are actually useful in practice? For more advanced machine learning methods (in particular, the methods you address in Section 2.4), could this question be addressed via simulation?

I’m confused by BLUP and it’s variants: having the mathematical definition here would be helpful: “Essentially, it is a (Bayesian) ridge regression model.” is not sufficient. Is it Bayesian, or not, and how does that differ from traditional regression and ridge regression?

---

### Review · Reviewer_Bmzr · 2023-09-19

**Summary Of Contributions:**

This paper provides a kind of survey for machine learning-based prediction of quantitative traits from genetic information. The authors overview models, techniques, datasets, evaluation metrics, etc., for genetic prediction. Specifically, this paper focuses on height prediction from genetic information and explains several datasets, techniques, and things to be considered in model development.

**Audience:**

Yes

**Broader Impact Concerns:**

This paper does not include the Broader Impact Statement. As the datasets explained in this paper might include genetic and personal information, it might be better to add the statement of ethical considerations.

**Claims And Evidence:**

Yes

**Requested Changes:**

- The authors should clarify the motivation and contribution of this paper.
- The inherent difficulty of phenotype prediction from genetic information compared to other machine learning tasks, such as bioinformatics tasks, should be clarified.
- As this paper is a kind of survey paper, the novelty and contribution compared to existing ones should be clarified.

**Strengths And Weaknesses:**

[Strengths]
- This paper provides a guide and introduction to prediction tasks from genetic information from the viewpoint of genetics, which might be useful for some machine learning researchers.

[Weaknesses]
- The techniques explained in this paper seem very common and well-known in the community.
- The motivation and contribution of this paper are not well-presented.
- The novelty compared to existing survey papers and guides is unclear.
- This kind of survey or guidance paper might be out-of-scope of TMLR. (https://jmlr.org/tmlr/editorial-policies.html)

---

### Note · Authors · 2023-09-28

**Comment:**

Dear Reviewers, Dear Editor,

On behalf of all co-authors, I would like to thank you for your helpful comments, which we largely included into our manuscript. Nonetheless, after careful considerations and based on the reviews received, we reconsidered the fit of our work with TMLR and decided to retract our manuscript.

Thank you again for your time and feedback.

**Withdrawal Confirmation:**

I have read and agree with the venue's withdrawal policy on behalf of myself and my co-authors.